# Colorectal Cancer: Therapeutic Approaches and Their Complications

**DOI:** 10.3390/biomedicines13071646

**Published:** 2025-07-05

**Authors:** Adebisi Adeleke, Amusa S. Adebayo, Kafilat Agbaje, Oluwabukunmi Olajubutu, Simeon K. Adesina

**Affiliations:** College of Pharmacy, Howard University, 2400 6th St NW, Washington, DC 20059, USA; adebisi.adeleke@bison.howard.edu (A.A.); kafilat.agbaje@bison.howard.edu (K.A.); oluwabukunmi.olajub@bison.howard.edu (O.O.); simeon.adesina@howard.edu (S.K.A.)

**Keywords:** colorectal cancer, therapeutic approaches, complications

## Abstract

Colorectal cancer (CRC) is ranked as the third most lethal of all cancers in the USA, following prostate and lung malignancy in men, and breast and lung malignancy in women, respectively. The risk factors for developing colorectal cancer fall into two categories: modifiable risk factors (obesity and physical inactivity, diet, smoking, alcohol, medications, diabetes, and insulin resistance) and non-modifiable risk factors (race and ethnicity, sex, age, and inflammatory bowel disease). The standard therapeutic approaches to the treatment of colorectal cancer have led to a reduction in the burden of colorectal cancer in the USA, with national statistics revealing a reduction in both the incidence and death rates. At the same time, five-year survival rates have also greatly improved. However, associated with these standard treatments are complications, which have become a burden (physical and emotional, financial, and economic burdens, and disability-adjusted life years), affecting the quality of life of CRC patients. This paper discusses the standard therapeutic approaches to managing colorectal cancer, the associated complications, and their management. In addition, a summary of the newly introduced therapeutic approaches for treating CRC, reported improvement in effectiveness over existing strategies and corresponding reduction in therapeutic complications will be discussed.

## 1. Introduction

Colorectal cancers are slowly developing cancers that can start as tumors or tissue growth on the inner linings (glandular or epithelial cells) of the rectum or colon to form polyps [1,2]. The polyp can eventually transform into cancerous entities by forming a tumor on the walls of the rectum or colon. This transformation occurs when the cells of the epithelium experience genetic mutations that confer a selective advantage on them [3,4]. Subsequently, the tumor then grows into blood vessels or lymph vessels, increasing the chances of metastasis to other anatomical sites [1].

The majority (95%) of the cancers that begin in the colorectal region are classified as adenocarcinomas [1,2]. Adenocarcinoma erupts in the mucus-secreting glands that line the colon and rectum [1,2]. The other types that are less prevalent include carcinoid tumors (erupt from hormone-producing intestinal cells), gastrointestinal stromal tumors (erupt from interstitial cells of Cajal), lymphomas (cancer of the immune systems forming in the colon or rectum), and sarcomas (normally starts from blood vessels; however, occasionally formed in the colorectal walls) [1,2,5].

Colorectal cancer has been ranked as the third deadliest of all cancers in the USA, following prostate cancer and lung cancer in males, and breast cancer and lung cancer in females (8% of all new cases) [1,6]. CRC is also recognized as the second most costly cancer to treat, accounting for 12.6% of all cancer treatment costs [7]. The total annual medical cost is estimated at $24.3 billion, with $23.7 billion spent on medical services and $0.6 billion spent on prescription drugs [7]. The risk factors for CRC have been broadly categorized into two groups: non-modifiable risk factors and modifiable risk factors. The non-modifiable risk factors include race and ethnicity, sex, age, and inflammatory bowel disease (IBD). In the US, CRC is more common among African Americans and Native Americans, and they suffer lower survival rates among all stages of CRC [3]. Based on sex, males have a higher chance (about 1.5-fold) of developing CRC than females across all ages and nations [3,8]. Age greatly influences the incidence of CRC. In the US, people over 65 years old are about three times more likely to be diagnosed with CRC than people aged 50–64, and a likelihood of 30 times more to be diagnosed than people aged 25–49 years old [3]. Modifiable risk factors include obesity and physical inactivity, diet, smoking, alcohol, medications, diabetes, and insulin resistance [3].

The standard treatment involves surgery combined with radiotherapy and/or chemotherapy, which is highly dependent on the site of the tumor and the progression of the disease [9]. These therapeutic approaches have led to a reduction in the burden of colorectal cancer in the USA, with national statistics revealing a reduction both in the incidence and death rates. At the same time, five-year survival rates have greatly improved [1]. However, associated with these standard treatments are complications, which have become a burden that affects the quality of life of CRC patients.

The burden associated with complications of CRC treatment includes physical and emotional, financial, and economic burdens, and disability-adjusted life years [10,11]. In a meta-synthesis study by Rutherford et al., from the patient-reported outcomes and experiences from the perspective of colorectal cancer survivors, it was shown that stoma problems seen in CRC survivors after treatment impair their physical, social, sexual, and psychological functioning of the patients [11]. Furthermore, according to the work of Regenbogen et al. (2014), patients who reported complications after surgery had a composite financial burden, which makes them more likely to spend their savings, borrow or take loans, fail to pay their credits, have a reduction in the money spent on food and clothing, decrease recreational activities, have concern about their finances, and took longer time to return to work [10]. Thus, the financial stress associated with CRC complications has a significant impact on survivors’ quality of life.

Therefore, therapeutic complications of CRC treatment and the associated burdens necessitate the need for developing strategies for their management to improve the quality of life of the patient. The ultimate goal is to develop newer therapeutic approaches with fewer complications and burden that would improve the compliance of CRC patients with the treatment regimen, increase the survival rates, and enhance the overall quality of life of the patients after treatment. The newer therapeutic approaches to achieving these goals include targeted therapy, immunotherapy, and combination therapies, among many others [12,13]. Thus, this paper discusses the standard therapeutic approaches to managing colorectal cancer, the associated complications, and the strategies for managing the complications. In addition, the newer therapeutic approaches developed towards treating CRC, with an improvement over the conventional treatment approaches in effectiveness and with reduced complications, will be discussed.

## 2. Research Method

**Search strategy:** Google Chrome engine; database: PubMed; search terms: COLORECTAL CANCER, THERAPEUTIC APPROACHES, therapeutic COMPLICATIONS.


**Returned MESH:**


Search: COLORECTAL CANCER, THERAPEUTIC APPROACHES, THERAPEUTIC COMPLICATIONS.

(“colorectal neoplasms OR (“colorectal”[All Fields] AND “neoplasms”[All Fields]) OR “colorectal neoplasms”[All Fields] OR (“colorectal”[All Fields] AND “cancer”[All Fields]) OR “colorectal cancer”[All Fields]) AND (“therapeutical”[All Fields] OR “therapeutically”[All Fields] OR “therapeuticals”[All Fields] OR “therapeutics”[MeSH Terms] OR “therapeutics”[All Fields] OR “therapeutic”[All Fields]) AND (“approach”[All Fields] OR “approach s”[All Fields] OR “approachability”[All Fields] OR “approachable”[All Fields] OR “approache”[All Fields] OR “approached”[All Fields] OR “approaches”[All Fields] OR “approaching”[All Fields] OR “approachs”[All Fields]) AND (“therapeutical”[All Fields] OR “therapeutically”[All Fields] OR “therapeuticals”[All Fields] OR “therapeutics”[MeSH Terms] OR “therapeutics”[All Fields] OR “therapeutic”[All Fields]) AND (“complicances”[All Fields] OR “complicate”[All Fields] OR “complicated”[All Fields] OR “complicates”[All Fields] OR “complicating”[All Fields] OR “complication”[All Fields] OR “complication s”[All Fields] OR “complications”[MeSH Subheading] OR “complications”[All Fields]).

## 3. Results

### 3.1. Search Terms and Articles Found in PubMed Search

Figure 1 shows the search strategy, number of articles, and the selection criteria for inclusion and exclusion.

### 3.2. Therapeutic Approaches in Colorectal Cancer Treatment

Figure 2 shows therapeutic approaches to colorectal cancer treatment. These include surgical intervention, radiotherapy, chemotherapy, targeted therapy, immunotherapy, and drug combination therapy.

### 3.3. Surgical Intervention

Surgical intervention is considered to be the most common treatment for CRC patients [14]. Surgical procedure options for CRC are dependent on the site where the tumor is found and the size. The available ones include laparoscopy, radiofrequency ablation, cryoablation, or colostomy [14]. Laparoscopy involves inserting different scopes by making smaller notches into the abdomen [14]. Polypectomy involves the surgical removal of the polyps during colonoscopy [12]. Surgery has been the mainstay curative treatment option in patients with non-metastasized CRC [15]. Surgical removal of the precancerous or cancerous tumor is an effective option and could allow for full recovery of the patient in cases of small, localized cancerous growths [12]. In addition, for solid tumors that have shown resistance to radiation and chemotherapy, surgery is required.

### 3.4. Radiotherapy

Radiotherapy is one of the mainstay treatment modalities of CRC alongside surgery, chemotherapy, and immunotherapy [16]. It involves the use of ionizing radiation in the management of proliferating tumorous cells [12]. The treatment utilizes ionizing radiation to induce DNA damage by breaking the double-stranded DNA, which leads to cell death [16]. It is used to treat all forms of cancer. However, because of the risk radiation poses to normal tissue, which includes cancers induced by radiation, radiation beams are often aimed at different angles of exposure to ensure the intersection at the tumor. This provides a larger absorbed dose of the radiation directly at the tumor site rather than the regular tissue sites. Different forms of radiation therapy have been reported to be used in the treatment of oligometastasis, which include selective internal radiotherapy (SIRT), trans-arterial chemoembolization (TACE), and radiofrequency ablation (RA) [12].

### 3.5. Chemotherapy

The treatment of CRC, both at the early stage and at the metastatic stage, is majorly centered on the use of chemotherapy [9]. This method involves destroying the tumor cells through cytotoxicity to the tumor cells which can lead to the suppression of the tumor [12]. The classes of drugs include agents such as alkylating agents, antimetabolites, plant alkaloids, and agents that influence biological responses that could either destroy the tumor cells, suppress the tumor growth, or suppress its cell division [12].

The chemotherapeutic agents that are being used in metastatic cases includes fluoropyrimidines (5-fluorouracil, 5-FU), oxaliplatin, and irinotecan. Aside from this, treatment also involves the use of multiple agent regimens that combine 5-FU with oxaliplatin (OX), capecitabine (CAP or XELODA or XEL), and irinotecan (IRI) [17]. The core of treatment for advanced CRC involves combining 5-FU and leucovorin with either oxaliplatin or irinotecan. The usage of the chemotherapeutic agent has resulted in a median overall survival ranging from 18 to 20 months [9,17,18].

#### 3.5.1. Fluoropyrimidines

##### Intravenous Fluorouracil

Fluorouracil (5-FU) is a synthetic fluorinated pyrimidine analog with a general activity through the inhibition of thymidylate synthetase, which is the rate-limiting enzyme in pyrimidine nucleotide synthesis [19,20]; thus, it inhibits DNA replication. This has been the mainstay of systemic treatment for colorectal cancer [21,22]. 5-FU is commonly administered with reduced folate, leucovorin, attributed to stabilizing fluorouracil’s interaction with the enzyme [23]. Ghoshal & Jacob (1997) [24] suggested that the anti-proliferative action of 5-FU may not be due exclusively to inhibition of DNA replication. They opined that the persistent inhibition of cellular proliferation following treatment of 5-FU-inhibited cells with exogenous thymidine is indicative of cytotoxicity at the RNA level [24]. Gorlick & Banerjee (2002) reported that on activation and downstream enzymatic processing of 5-FU into fluorodeoxyuridine mono-, di-, and triphosphates, the triphosphate is incorporated into nuclear and cytoplasmic NRA, causing cell apoptosis [25]. Inhibition of DNA or RNA is dependent on the rate of infusion, and the difference in actions has been attributed to the pharmacokinetic parameters, with the continuous infusion showing greater efficacy and activity against DNA compared to IV bolus, which showed greater activity on RNA [26,27]. In patients with metastatic cancer, the combination treatment has been shown to reduce the tumor size by 50% or more in about 20% of the patients, and the median survival increased from approximately 6 months to an average of 12 months [28].

##### Oral Fluoropyrimidines

Oral fluoropyrimidines experience erratic absorption when administered due to the varying concentrations of mucosal catabolic enzyme, dihydropyrimidine dehydrogenase [21,22]. However, there are two approaches developed to bypass the problem. The first process involves using an absorbable fluorouracil prodrug, which is not broken down by the catabolic enzyme, while the second step involves co-administering the drug with a dihydropyrimidine dehydrogenase inhibitor [29]. Examples include capecitabine, an oral prodrug of fluorouracil that undergoes a 3-conversion step to fluorouracil, while tegafur uracil escapes the erratic intestinal absorption of fluorouracil when they are co-administered with dihydropyrimidine dehydrogenase (uracil) inhibitor. This permits uniform absorption and bioavailability of the drug.

#### 3.5.2. Topoisomerase Inhibitors (Irinotecan)

Irinotecan is a semisynthetic derivative of a natural alkaloid, camptothecin, which is often transformed into SN-38 by carboxylesterases [30]. Sn-38 has been reported to cause fragmentation of DNA fragmentation and programmed cell death through the inhibition of topoisomerase I. The topoisomerase I enzyme catalyzes the breaking and rejoining of DNA strands using the replication of DNA [21]. It forms a topoisomerase-inhibitor-DNA complex that affects DNA function [20]. Thus, a higher concentration of Topoisomerase I makes the cell more sensitive to irinotecan [20,31]. Enzymes like carboxylesterases (CES), β-glucuronidase, uridine diphosphate glucuronosyltransferase (UGT), hepatic cytochrome P-450 enzymes CYP3A, and ATP-binding cassette (ABC) transporter protein play vital roles in the uptake and metabolism of irinotecan.

SN-38 is metabolized in the liver. It is rendered inactive by a polymorphism in the uridine diphosphate glucuronosyltransferase isoform 1A1 (*UGTA1A1*) gene through the glucuronidation process and eliminated through the biliary system. The gene causes a reduction in the inactivation of SN-38 followed by a subsequent increase in treatment-related toxicity [32,33,34]. The drug is not used in patients having hyperbilirubinemia because of the potential increase in the bilirubin level associated with irinotecan toxicity [21]. Other common toxic effects associated with the drug include diarrhea, myelosuppression, and alopecia.

A combination of irinotecan with either (FOLFIRI) infusion or (IFL) fluorouracil and leucovorin bolus in the preliminary treatments of patients having metastatic CRC (mCRC) has been shown to improve the progression-free and overall survival of patients [35,36,37].

#### 3.5.3. Platinum Compounds (Oxaliplatin)

Oxaliplatin is a diaminocyclohexane, a third-generation platinum compound that works by forming DNA adducts, leading to the impairment of DNA replication and cellular apoptosis [38]. The usage was approved in Europe in 1996, but the USA FDA granted accelerated approval in 2002, and full approval for use in combination with 5-FU granted in 2004 for advanced CRC or mCRC [39]. Administering singly in patients with mCRC has limited efficacy; however, when administered with fluorouracil and leucovorin, there is a significant clinical benefit observed. This has been linked to the possibility of oxaliplatin-induced down-regulation of thymidylate synthetase [21].

Oxaliplatin has a 1,2-diaminocyclohexane ligand (DACH), a unique feature in the structure that differentiates it from other platinum compounds. DACH makes it difficult for DNA repair, thereby improving the tumor cell-killing effect of oxaliplatin [20]. The toxic effect of oxaliplatin is manifested through cumulative sensory neuropathy, with features of paresthesia of the hands and feet.

Clinical studies have shown that adding oxaliplatin to the infusion of fluorouracil and leucovorin (FOLXOX) enhanced the tumor response rate and disease-free survival, having a trend that depicts an increase in overall survival [21].

#### 3.5.4. Antimetabolites (Capecitabine)

This is the earliest oral chemotherapeutic drug that was developed for treating CRC. When metabolized inside the system, it is changed to 5′-deoxy-5-fluorocytidine (5′-OFCR) and 5′-deoxy-5-fluorouridine (5′-DFUR). Thereafter, TP ultimately hydrolyzes 5′-DFUR to 5-FU, exerting its cytotoxic activities. The multinational phase III trial has provided evidence to support the combined therapy containing capecitabine and irinotecan (XELIRI), with or without bevacizumab, as a second-line treatment modality in patients having mCRC [20,40,41].

### 3.6. Targeted Therapy

Targeted therapies are known to work on cancerous cells through the inhibition of cell proliferation, differentiation, and migration. They can also alter the tumor microenvironment, including local blood vessels and immune cells, impeding tumor development and exerting robust surveillance and assault from the immune system [17]. Smaller molecules have been a major player in targeted therapies. This set of molecules, like monoclonal antibodies, because of their molecular weight (<900 Da), has a strong ability to penetrate cancerous cells [17]. They work within the cell to render inactive selected enzymes, which inhibit cancer cell growth and, in addition, prompt apoptosis. The molecular targets include cyclin-dependent kinases, proteasomes, and poly (ADP-ribose) polymerase [17].

In another way, targets outside the cell, including receptors on the cell surface or other sites bound to the membrane, can be recognized by monoclonal antibodies or therapeutic antibodies, which bind them to exert direct regulation of the downstream cell cycle progression and cell death. Furthermore, there is a report that certain monoclonal antibodies target other cells, different from tumor cells, like immune cells, that can influence the immune system to attack human cancer [17].

The pathways that facilitate initiation, progression, and migration of CRC, like Wnt/β-catenin, Notch, Hedgehog, and TGF-β (transforming growth factor-β)/SMAD, and, inclusively, those capable of activating signaling cascades, like phosphatidylinositol 3-kinase (PI3K)/AKT or RAS/rapidly accelerated fibrosarcoma (RAF), are potential sites for cancer targets and druggability [17].

Following chemotherapy, targeted therapies, like monoclonal antibodies and small-molecule inhibitors, have been an effective treatment option in CRC patients. Target therapy involves targeting specific genes and proteins, hindering the growth and survival of the cancer cells [14]. Targeting monoclonal antibodies to vascular endothelial growth factors (VEGF) and epidermal growth factor receptor (EGFR), enhanced the overall survival for CRC to three years [20]. They are known to exhibit lower side effects as compared to chemotherapy [20]. However, treatment with anti-EGFR is associated with high incidence of mucositis and, to a lesser degree, electrolyte imbalances, notably hypomagnesaemia. By far the most problematic side effects to manage are skin reactions, particularly papulopustular rash [42]. In addition, the chimeric mAb cetuximab is associated with enhanced risk of infusion reactions. Appropriate prophylaxis and patient-centered adverse effects management strategies that are tailored to the degree of toxicity should be implemented to ensure patient compliance with anti-cancer therapy regimen.

#### 3.6.1. Angiogenesis Inhibitors

Angiogenesis plays a great role in tumor growth and survival [39]. Hypoxia in the tumor microenvironment (TME) is associated with upregulating of hypoxia-inducible factor (HIF), followed by the induction of the production of vascular endothelial growth factor (VEGF) [40,43]. The overexpression of the VEGF gene and the high levels of circulating VEGF protein have been linked with a flawed prognosis in CRC [44]. Angiogenesis involves the establishment of new blood vessels by the cancer to feed the tumor cells with a nutrient and oxygen supply. Angiogenesis inhibition has been recognized as a strategy for controlling malignant proliferation and its spread [21]. The current focus in the antiangiogenic strategy involves the inhibition of vascular endothelial growth factor (VEGF) [21]. VEGF is a soluble protein that promotes blood vessel proliferation.

Bevacizumab is an antiangiogenic drug in this class, and the first to precisely target VEGF, leading to a reduction in tumor growth. It is a humanized monoclonal antibody used against VEGF, combined with chemotherapy in patients with advanced CRC [21]. Bevacizumab works through the reduction in the formation of new blood vessels needed by growing tumors for continuous development [39]. Though bevacizumab is well tolerated, it has a mild toxic effect of reversible hypertension and proteinuria. Rare cases of toxicity include bowel perforation, serious bleeding events, risk of arterial embolic events, and reversible posterior leukoencephalopathy syndrome [45,46,47]. Clinical studies show improvement in tumor response rate and progression-free survival when bevacizumab is administered with fluorouracil and leucovorin in patients with metastatic colorectal cancer. In addition, adding bevacizumab to FOLFIRI or FOLFOX in patients having untreated mCRC has established an enhanced rate of response and progression-free survival times [48,49].

#### 3.6.2. Epidermal Growth Factor Receptor (EGFR) Inhibitors

To inhibit the function of the EGFR, scientists have developed several antibodies targeting the extracellular domain of EGFR coupled with small molecular inhibitors of the intracellular tyrosine kinase domain. Only anti-EGFR monoclonal antibodies, cetuximab and panitumumab, showed efficacy in the treatment of CRC [50]. As a chimeric immunoglobulin G (IgG), cetuximab works by inducing internalization of EGFR and inducing degradation after it has bound to the external domain of EGFR. Binding to the external domain of the EGFR prevents ligand binding, which prevents cell growth and survival [39]. The binding brings about receptor internalization and degradation without activation or phosphorylation [39,51]. In addition, it has also been established that the binding of cetuximab to the receptor induces antibody-mediated cytotoxicity, progressing to tumoral cell death [39,52]. Furthermore, a study shows that cetuximab down-regulated VEGF expression, therefore lowering tumor angiogenesis [53]. Cetuximab has been shown to improve progression-free survival (PFS) in patients who have previously shown poor response to singular-agent irinotecan therapy in metastatic cancer [17].

When patients who were having disease progression while on fluoropyrimidine, irinotecan, and oxaliplatin were changed to weekly cetuximab administration, an improvement in progression-free and overall survival, compared to those with supportive care treatment alone, was observed [21,54]. In a further study with patients with irinotecan-refractory metastatic cancer, the use of cetuximab alone gave a tumor response rate of 10%, while a response of 20% was reported when cetuximab was used with irinotecan [55]. This established cetuximab’s ability to overcome tumor cells with irinotecan resistance. Other studies also indicated there is a prolonged overall survival (OS) and PFS when cetuximab is used in patients with CRCs that have either experienced failure with treatment using fluoropyrimidine (IRI and OX) or where it is contraindicated [17].

Panitumumab is a humanized monoclonal antibody also used in the targeting of EGFR, but with biweekly dosing, with similar activity to cetuximab in mCRC [50]. There was a positive tumor response in patients (9%) who had earlier been treated with fluorouracil combined with either oxaliplatin or irinotecan, after being treated with panitumumab [56]. Due to the type of patients who respond to the use of cetuximab and panitumumab, molecular markers have been used to predict tumor response when investigated. This can be used to define patients’ subsets who can have great benefits if treated with EGFR inhibitors [21]. Two tumor characteristics are presently in use for this purpose: EGFR copy number (as established using fluorescence in situ hybridization) and K-ras gene mutation stage [21]. A high EGFR copy number using fluorescence in situ hybridization has a link with an increasing rate of tumor response and prolongation of disease-free and overall survival. On the other way around, patients having a mutation in K-ras have developed resistance when treated with cetuximab or panitumumab [57], with reduced response rates and poorer survival. Regorafenib (Fluror-sorafenib, stivarga), a small-molecule inhibitor that targets multiple tyrosine kinases, was approved by FDA for CRC treatment [58]. It has a chemical structure that is closely related to sorafenib and is the first approved pharmacotherapy for mCRC patients who are non-responsive to current standard therapies. 

### 3.7. Immunotherapy (Immune Checkpoint Inhibitors)

Immunotherapy has been considered to be the fourth treatment modality in CRC treatment, based on advances in molecular biology, cell biology, and immunology [12]. Immunotherapy involves the use of immune checkpoint inhibitors (ICIs) to enhance the patient’s immune system’s capability to adequately identify and overcome cancer cells [59]. This strategy involves interrupting immunosuppressive signals within the TME and reactivating the antitumor immunity through the targeting of the molecular immune checkpoints [60]. Tumors create microenvironments that are immunosuppressive, which allows them to escape immune system surveillance. This is achieved with the presence of immunosuppressive tumor-associated macrophages, myeloid-derived suppressor cells, and regulatory T cells residing within the tumor. The products of these results in a microenvironment that disrupts the activation of the immune system activation and any attack [12,61]. The ICIs modulate the interaction of T cells, antigen-presenting cells (APCs), and tumor cells, aiding the activation of the suppressed immune responses [62]. This has been found effective for patients with mCRC that is mismatch-repair-deficient (dMMR) or microsatellite instability-high (MSI-H) [62]. The clinical study of programmed cell death receptor (PD)-1 mAb in patients having mCRC revealed promising effects. The progression-free survival rates of 78% and 11% were exhibited in mismatch-repair-deficient (dMMR) and mismatch-repair-proficient (pMMR) tumors, respectively [63,64].

The dMMR tumors have a high mutational burden, tumor-infiltrating lymphocytes (TILs) enrichment, and upregulated PD-L1 expression within the TME [65]. These features enable the CRC in this category to have better responses to immune checkpoint immunotherapies [60,64]. In 2017, the US Food and Drug Administration (FDA) approved pembrolizumab (anti-PD-1) usage for advanced/metastatic solid malignancies with dMMR or MSI, including use in CRCs [60]. In addition, nivolumab (with or without ipilimumab) was also approved to serve the same purpose [62].

However, it is imperative to note that dMMR or microsatellite unstable (MSI) CRC tumors is just 14% of all CRCs, which is a smaller percentage [60,63]. The mCRC is categorized by inadequate mutated tumor antigens. This portends a great challenge in immunotherapy, providing benefits for a lot of mCRC patients who have mismatch-repair-proficient (pMMR) or microsatellite-stable (MSS) or low microsatellite instability (MSI-L) (termed pMMR/MSS/MSI-L mCRC [62]. To date, the only predictor biomarker for response to checkpoint inhibitors is the presence of dMMR in the cancer tumor.

### 3.8. Combination Therapies Approach for Colorectal Cancer Treatment

Combining two or more therapeutic treatments to specifically target cancer-inducing or cell-sustaining pathways is a fundamental strategy in cancer therapy [66,67]. While monotherapy remains a commonly employed approach for various types of cancer, it is generally regarded as less effective compared to combination therapy. Traditional monotherapeutic methods indiscriminately attack rapidly dividing cells, leading to the destruction of both healthy and cancerous cells [68]. Chemotherapy, a frequently used monotherapy, can cause significant toxicity, with numerous side effects and risks. It often weakens the immune system by damaging bone marrow cells, thereby increasing vulnerability to infections and other diseases [69,70].

Although combination therapy involving chemotherapeutic agents may also present toxicity risks, these are typically reduced because multiple pathways are targeted. This synergistic or additive effect allows for lower therapeutic doses of each drug [71]. Furthermore, combination therapy has the potential to minimize toxic effects on healthy cells while maintaining cytotoxic efficacy against cancer cells. This is possible when one drug in the regimen antagonizes the cytotoxic effects of another drug on normal cells, effectively shielding healthy cells while still attacking cancerous ones [72].

#### 3.8.1. FOLFIRI

FOLFIRI is one of the most widely used standard chemotherapy regimens for colorectal cancer (CRC) and consists of three drugs: irinotecan, 5-fluorouracil (5-FU), and leucovorin. 5-FU acts as a pyrimidine antagonist. Its activation begins with its conversion by orotate phosphoribosyltransferase (OPRT) and uridine phosphorylase (UP) into fluorouridine monophosphate (FUMP) and fluorouridine (FUR). FUR is subsequently converted to FUMP via uridine kinase (UK), which is then phosphorylated into fluorouridine diphosphate (FUDP) and further into the active metabolites fluorouridine triphosphate (FUTP) or fluorodeoxyuridine diphosphate (FdUDP) through ribonucleotide reductase (RNR) FUTP, a fluorinated RNA nucleotide analog, can be mistakenly incorporated into tumor cell RNA, causing RNA damage. Meanwhile, FdUDP can be phosphorylated or dephosphorylated to produce active metabolites such as fluorodeoxyuridine triphosphate (FdUTP) and fluorodeoxyuridine monophosphate (FdUMP) [73].

The FOLFIRI regimen is typically administered every two weeks across multiple treatment cycles. However, patients often develop congenital or acquired resistance to 5-FU, which limits its efficacy as a monotherapy and poses a significant challenge to effective clinical treatment [74]. To address this, 5-FU is usually administered with leucovorin, which enhances the drug’s binding affinity to thymidylate synthase (TS), thereby improving its efficacy [75]. A meta-analysis confirmed these findings, showing that combining 5-FU with leucovorin increases response rates (RR) and overall survival (OS) compared to 5-FU alone [28].

The addition of irinotecan to the regimen further enhances its effectiveness. Compared to 5-FU and leucovorin alone, the FOLFIRI regimen demonstrates significant improvements in progression-free survival (PFS), OS, and RR, while effectively delaying cancer progression [76]. Importantly, irinotecan does not exhibit cross-resistance with 5-FU/leucovorin therapy, making it a valuable component of combination therapy for CRC.

#### 3.8.2. FOLFOXIRI

The FOLFOXIRI regimen, combining irinotecan, oxaliplatin, and 5-FU/leucovorin, is a high-intensity chemotherapy protocol used for colorectal cancer (CRC). Oxaliplatin, one of the cytotoxic agents in this regimen, primarily targets DNA, although its exact mechanism of action remains incompletely understood. Studies indicate that oxaliplatin induces cytotoxicity mainly through DNA damage. Upon entering tumor cells, oxaliplatin forms platinum-DNA adducts, disrupting DNA transcription and replication, ultimately leading to tumor cell death [76,77]. The antitumor process of oxaliplatin involves four key phases: drug uptake, activation through hydration, DNA platinization, and intracellular processing.

The FOLFOXIRI regimen has demonstrated significant efficacy in improving objective response rates (RR), progression-free survival (PFS), and overall survival (OS), with manageable and well-tolerated side effects. It is now recommended by the Chinese Society of Clinical Oncology (CSCO), the European Society for Medical Oncology (ESMO), and the National Comprehensive Cancer Network (NCCN) for treating advanced CRC. Early clinical evidence supported the feasibility and effectiveness of combining irinotecan and oxaliplatin for CRC [78,79]. FOLFOXIRI’s safety and efficacy as a first-line treatment for metastatic CRC (mCRC) were first reported in 2002 [80].

The HORG study, which included 283 patients, reported higher incidences of alopecia, diarrhea, and neurotoxicity in the FOLFOXIRI group compared to FOLFIRI. While OS improved in the FOLFOXIRI group (21.5 vs. 19.5 months), the difference was not statistically significant (*p* = 0.337) [81]. Additionally, the tolerability of FOLFOXIRI is poorer among Asian populations compared to Europeans, leading to modifications in its application in China. For example, the irinotecan dosage in FOLFOXIRI was reduced from 180 mg/m^2^ to 150–165 mg/m^2^ to suit Chinese patients, although further studies are required to confirm the efficacy and safety of this modified regimen [82].

Despite these challenges, FOLFOXIRI has shown promising results, including a PFS of 13.37 ± 9 months and an overall response rate of 79.4%, while maintaining side effects within an acceptable range for mCRC first-line therapy [83].

#### 3.8.3. XELIRI

The XELIRI regimen, which combines irinotecan with capecitabine, offers a more convenient chemotherapy option, requiring only 2–3 h of infusion every three weeks. A phase II single-arm study demonstrated favorable efficacy and safety in patients with metastatic colorectal cancer (mCRC) [84]. Capecitabine, an oral fluorouracil prodrug with nearly 100% bioavailability, is notable for its convenience, safety, and significant antitumor activity [84]. After oral administration, capecitabine is absorbed through the intestinal mucosa and metabolized in the liver by carboxylesterase (CES) into 5′-deoxy-5-fluorocytidine (5′-DFCR) [85]. It is then converted into 5′-deoxy-fluorouracil (5′-DFUR) by cytidine deaminase (CD), which is abundant in both liver and tumor tissues [86]. The final step occurs predominantly in tumor tissues, where thymidine phosphorylase (TP) converts 5′-DFUR into 5-FU, enhancing antitumor effects while minimizing systemic toxicity due to low TP activity in normal tissues.

A meta-analysis comparing XELIRI and FOLFIRI regimens for first-line mCRC therapy found no significant differences in overall survival (OS), response rates, or progression-free survival (PFS), and their safety profiles were comparable [87]. However, the XELIRI regimen offers greater convenience due to capecitabine’s oral administration, reducing the need for prolonged infusions and central venous access required by FOLFIRI [88].

Despite these advantages, XELIRI is associated with increased gastrointestinal toxicity compared to FOLFIRI [48,89]. To address this, the AXEPT trial, a large multicenter randomized phase III study, evaluated a modified XELIRI (mXELIRI) regimen. The study demonstrated that mXELIRI, with or without bevacizumab, is a viable alternative to FOLFIRI as a second-line treatment for mCRC, with comparable OS (16.8 vs. 15.4 months, *p* < 0.0001) and a significantly lower incidence of grade 3–4 neutropenia (17% vs. 43%) [90].

From a cost-benefit perspective, mXELIRI has been found to be a cost-effective alternative for second-line mCRC treatment, and it is also a more tolerable and convenient option than FOLFIRI [91].

### 3.9. Personalized Medicine

Although the use of chemotherapy is widespread in the treatment of colorectal cancer, several challenges have limited the degree of success obtained with their use. Major challenges include a lack of specificity, resistance development, and tumor heterogeneity in patients. Efforts towards mitigating these challenges are currently being directed at personalized medicine and targeted therapy. Personalized medicine involves the use of drugs that specifically target genes or pathways necessary for the proliferation of the cancer cells in individual patients. Before this can be applied, adequate genetic profiling is required to identify potential genes, biomarkers, targets or pathways that can be targeted in the patients.

Several biomarkers that can be targeted have been identified, of which the KRAS and BRAF genes are popular targetable biomarkers in the treatment of CRC. Studies have shown that BRAF mutations are found in roughly 10–15%, while KRAS mutations are found in about 30–40% of diagnosed colorectal cancer cases [92]. The occurrence of these two genetic mutations, previously thought to be mutually exclusive, has now been shown to occur concurrently [93]. Presently, BRAF and KRAS mutations are significant biomarkers in colorectal cancer classification, primarily because they are reliable predictors of treatment response and prognosis, particularly for anti-EGFR therapy. Mutations in these genes, especially BRAFV600E, are associated with resistance to anti-EGFR agents and can impact overall survival, especially in specific subtypes of colorectal cancer [94].

BRAF inhibitors like encorafenib (Braftovi) and vemurafenib (Zelboraf) are used clinically in colorectal cancer. These inhibitors target the BRAF protein, a key player in cell growth, of which inhibition can help reduce cancer cell growth [95]. Encorafenib is often used in combination with other targeted therapies like cetuximab (Erbitux) and chemotherapy regimens (like mFOLFOX6). The BEACON CRC trial confirmed Encorafenib and cetuximab as the new standard of care for this patient population [96].

In clinical practice, sotorasib (Lumakras) and adagrasib (Krazati) are KRAS G12C inhibitors used in the treatment of advanced colorectal cancer with the KRAS G12C mutation. These drugs are often combined with EGFR inhibitors like cetuximab or panitumumab [97].

Clinical trials have demonstrated that the combination of KRAS G12C inhibitors with EGFR inhibitors results in longer progression-free survival (PFS: the time period during and after treatment when a patient lives without the cancer worsening) and improved overall survival in patients with KRAS G12C-mutated colorectal cancer compared to standard treatments [98].

## 4. Complications Associated with the Therapeutic Approaches and Their Management

### 4.1. Complications Associated with Surgery, and Management

Due to the complexities associated with colorectal cancer surgery, several complications have been reported in the literature. These complications include adhesion and small bowel obstructions (SBO), thrombosis, infections, and urogenital dysfunction (Figure 3) [99,100,101]. Other complications associated with colorectal cancer surgery include port site metastases, commonly seen in minimal resection of colorectal surgery [102], anastomotic leakage, and ileus that cause nausea/vomiting and pain [99,103], and colonic ischemia (ischemic colitis) [99].

#### 4.1.1. Adhesion and Small Bowel Obstructions (SBO)

Adhesions, based on reports, are a complication frequently linked with laparoscopy, with occurrence in about 95% of cases, and often the leading origin of small bowel obstruction [99]. It has been reported that about 10% of colorectal surgery, postoperatively, leads to SBO together with peritoneal adhesions [99], with a study showing that laparoscopic and open surgery have an equal association in the development of SBO [104]. The adhesive SBO recurrence leads to a reduction in the survival rate, thus, necessitating timely surgical management to suppress the recurrence [105]. Management of SBO includes non-surgical management and laparoscopic adhesiolysis [99]. Non-surgical management is achieved first by bowel decompression; nonetheless, the majority of patients need surgery [99]. On the other hand, laparoscopic adhesiolysis has been reported to achieve lower mortality and faster recovery [106], though with a report of a higher recurrence rate [107]. The preventive measures for SBO include the use of carboxymethyl cellulose and hyaluronic acid, and the use of poly(L-lactide-*co*-d,l-lactide) adhesion barrier to prevent the formation of peristomal adhesion [102,103,108].

#### 4.1.2. Thrombosis

The occurrence of thromboembolic events in CRC surgery has been associated with 2.5% of the cases [99]. There is a higher risk of this event in patients with the use of steroids, preoperative sepsis, weight loss history, extended surgical duration, and postoperative chemotherapy [100,109]. Both laparoscopic and open colorectal surgery have been associated with this event [110]. Preoperative screening has been reported to reduce intra- and postoperative complications [111]. In addition, the use of anti-thrombotic drugs for prophylaxis has been suggested to prevent thrombosis, for example, using low molecular weight heparin (LMWH) [112,113]. Furthermore, extended thromboprophylaxis perioperative, for a period of 30 days, has also been reported to cause reduction in thromboembolic events, contrary to 10 days of standard LMWH therapy [114].

#### 4.1.3. Infections

Postoperative infections have been reported to contribute to morbidity and mortality associated with colorectal cancer surgery [100]. Advanced age, perioperative complications, types of surgical wounds, and surgeries for neoplasms have been recognized as the major factors leading to the incidence of infections, while other factors include diabetes mellitus, chemotherapy, and the use of steroids [100]. Recommendations for preventing infections include mechanical bowel preparations (MBP) to clear the bowel in order to prevent sepsis, and MBP in combination with antibiotics [109,115].

#### 4.1.4. Urogenital Dysfunction

Urogenital dysfunction remains a frequent challenge occurring after colorectal cancer treatment, with surgery, radiation, or chemotherapy [101,104]. It has been reported that approximately more than half of patients who have undergone colorectal cancer treatment experience a decline in sexual function [116]. In males, sexual dysfunction after colorectal cancer includes ejaculatory dysfunction and impotence, and for women, commonly including vaginal dryness and dyspareunia [117]. It has been reported that about 63% of men develop ejaculation dysfunction, and about 63% have difficulty ejaculating after rectal cancer treatment [118,119]. It is also reported that urinary dysfunction arises in one-third of patients who have undergone treatment for rectal cancer [116]. Urinary and sexual dysfunction in colorectal cancer treatment has been linked to surgical nerve damage in low rectal cancer and abdominoperineal resection, while radiotherapy was linked to the progression of sexual dysfunction without necessarily affecting urinary function [117]. Surgical treatment could affect the parasympathetic and sympathetic nerves that are involved in erection and ejaculation [120], while radiation could impact nerves and blood vessels involved in erection [117].

In a study conducted by Perry et al. on sexual dysfunction resulting from surgery for rectal cancer based on a single-institution experience, it was reported that out of 147 patients examined, the overall sexual dysfunction rate was found to be 70% at a 38-month median time from surgery [121]. A total of 62% of men and 87% of women reported an overall score that fell below one standard deviation of the population mean. The study showed a high rate of the occurrence of sexual dysfunction after rectal cancer surgery, with more risk peculiar to female patients. In another study by Mannaerts et al., it was reported that of 73 men who had undergone both radiation and surgery as treatments for rectal cancer, about 10% could experience ‘quality erection’ and only about 10% could ejaculate postoperatively [119]. Sexual dysfunction could impact the quality of health experienced by colorectal cancer patients after treatment. Men who experience sexual dysfunction could experience depression, distress, and other psychological events that could impair their intimacy in relationships [122,123].

The management of sexual dysfunction involves the use of phosphodiesterase-5 inhibitors (PDE5i) and intracavernosal injections [124]. The mechanism of PDE5i revolves around increasing intracellular cGMP levels that allow for prolonged erection [125]. The PDE5i drugs approved by the FDA for erectile dysfunction include sildenafil, tadalafil, avanafil, and vardenafil [125,126,127]. PDE5i both stimulates libido and may result in better erectile experience [128]. Intracavernosal injections have also been used in the treatment of erectile dysfunction in males [126]. The most common of these is intracavernosal alprostadil which is a synthetic analog of prostaglandin E1 (PGE1). Alprostadil binds to G-coupled PGE1 receptors available on the surface of smooth muscle cells [129]. However, it is important that the physician ask the patient first, and consider alternatives for those who might have a fear of penile injections [124]. PDE5i and intracavernosal injections have been found to be efficacious and well-endured among patients [130]. The other option for males is the use of an inflatable penile prosthesis; however, this is faced with issues regarding reservoir placement.

The challenge with ejaculatory function, retrograde ejaculation has been managed with electroejaculation and alkalization of urinary pH to preserve sperm viability [131,132]. Anejaculation, a condition where a male finds it difficult to ejaculate any semen during the course of sexual activity associated with nerve damage, pharmaceutical side effects, or previous surgery, can be addressed using penile vibratory stimulation (PVS), though not the best option for sensate patients [133]. Electroejaculation is preferred in sensate patients [124]. In addition to this, dopaminergic drugs like oxytocin and clomiphene citrate, are other alternatives that have been used to enhance ejaculatory nerve sensitivity or semen production [134]. Even though several treatments for erectile dysfunction have been postulated, only about 50–80% of men comply with the medical interventions and only about 38% of such men found such treatment helpful in mending their sex lives [101,135].

### 4.2. Complications Associated with Radiotherapy and Management

Every year, several thousands of patients receive radiotherapy as part of their colorectal cancer treatment plan. While there have been some successes recorded with the use of radiotherapy, several patients have come down with mild to severe complications due to chemotherapy [136,137]. Jairam et al. [138] conducted a retrospective cohort study between January 2006 and December 2015; they found that about 1.5 million patients developed one or more complications associated with radiotherapy. The reported complications include, but are not limited to, neutropenia, sepsis, anemia, pneumonia, and acute kidney injury (Figure 4).

Intraoperative radiation therapy (IORT) has been shown to improve disease outcomes in advanced and recurrent CRC. However, it is associated with several short-term complications such as abscess fistulae, wound, and anastomotic leakage; and long-term complications including ureteric obstruction and sacral necrosis [139]. Most side effects gradually disappear in the weeks or months after treatment. However, some side effects can continue and might be noticed months or years later.

Chronic enteritis caused by radiation is usually managed as the symptoms arise. Diet modification is usually considered first, and it might take a period of time for a perfect dietary plan to be developed. For complications that present as diarrhea and urgency, patients are usually advised to avoid foods that are high in fiber [140,141]. Limiting lactose and dairy intake has also been found to be helpful. With the emerging interest in the involvement of gut microbiome in colorectal cancer, the use of probiotics in managing complications has also been considered. A meta-analysis of recent trials has suggested that probiotics, either in pill or drink form, are effective in managing radiation-induced diarrhea [142,143].

Another important complication that has attracted a lot of attention is radiation-induced infertility and sexual toxicity. This is increasingly becoming important to consider because of the increase in the number of young people being diagnosed with CRC. In female patients, unfortunately, the ovaries have been shown to be highly sensitive to radiotherapy, and the doses at which radiation is used for peri-operative or definitive purposes can result in infertility [144,145,146]. The risk is more common among women receiving radiotherapy for treatment of rectal cancer, compared to colon cancer, because this region is closer to the ovaries [147]. Due to this, women are usually advised to harvest their eggs prior to the start of radiotherapy. In the case of men receiving radiotherapy, the risk of infertility is approximately 20%. Therefore, sperm banking is recommended to maximize the probability of having biological children [148,149]. Conversely, for those who do not wish to have more children, contraception is advised during and after radiation therapy.

### 4.3. Mechanisms Underlying Radiation-Induced Sexual Toxicity

In men, sperm cells are made up of several subpopulations, including spermatozoa and spermatocytes. These subpopulations have varying degrees of sensitivity to radiation depending on their chromatin composition. When these cells are hit by radiation, they rely on repair proteins to regain their integrity. However, failure to utilize the repair proteins results in DNA damage and mutations, which further affect the quality and quantity of sperm cell production [149].

In female patients receiving radiotherapy, the uterus and ovaries are usually affected, and sensitivities of these important reproductive organs have been shown to vary with age. While the uterus is more sensitive to radiation therapy at a younger age, the sensitivity of the ovaries to radiation increases with age [150]. Although the number of oocytes produced between birth and menopause decreases naturally, patients receiving radiotherapy tend to have an accelerated decline in production [151]. Generally, immature, actively dividing and undifferentiated cells, e.g., stomach mucosa, basal skin layer, and stem cells, are more radio-sensitive while mature, differentiated, and non-actively dividing cells like neurons are more radioresistant [152]. Thus, cells with high mitotic activity and active DNA replication are more radiosensitive, whereas cells with low mitotic division rates are more radioresistant. However, female germ cells are an exception. Although progenitor female germ cells stop at the first meiotic division, they are extremely sensitive to radiation, which causes DNA damage [153].

### 4.4. Complications Associated with Chemotherapy and Management

Complications associated with drug interventions in colorectal cancer treatment are shown in Figure 5. The chemotherapeutic drugs used in the adjuvant treatment of colorectal cancer are known to produce undesirable neurological and gastrointestinal side effects [154]. Most often, the chronic side effects lead to dose limitations, while in severe cases, there is a need to stop the chemotherapy treatment because of the inability of the patient to tolerate the adverse effects, thus impeding the efficient treatment of CRC using chemotherapeutic agents [154,155]. Approximately 40% of patients placed on standard and high-dose chemotherapy experience pain, bloating, ulceration, vomiting, and diarrhea [156].

#### 4.4.1. Chemotherapy-Induced Diarrhea (CID)

Chemotherapy-induced diarrhea (CID) is one of the problematic frequent dose-limiting side effects associated with chemotherapeutic drugs which affects about 80% of patients placed on chemotherapy for colorectal cancer and other gastrointestinal cancers [157,158]. In addition, about 5% of early deaths experienced from chemotherapeutic combination treatment stem from CID.

CID in colorectal cancer patients is linked with the use of 5-FU, irinotecan, capecitabine, and oxaliplatin [54,159,160]. However, the incidence and the severity of CID differ depending on the chemotherapeutic drug and the combination, with the 5-FU and irinotecan combination having a high rate of about 87% [158,161]. The occurrence and persistence of CID in patients on combination therapy often led to malnutrition and dehydration, which often culminate in other secondary effects like weight loss, renal failure, fatigue, and hemorrhoids [162,163]. In addition, CID could cause severe inflammation and bowel wall thickening, ulceration [154,164]. Thus, CID has played a significant role in the disruption of clinical outcomes and treatment alterations. About 60% of treatment alterations arise from CID, having about 22% of patients having dose reductions, 28% of patients ending up with dose delays, and 15% of patients having treatment cessation due to severe diarrhea during CRC treatment [165,166].

The mechanisms underlying CID are not well understood. The most notable explanation in the literature is that it is a form of mucositis or its by-product [156]. Mucositis is an inflammation and ulceration of the mucous membrane of the gastrointestinal tract. GIT mucositis arises from the disruption of the intestinal microflora and mucin secretion, which stimulates the development of CID [156]. The toxicity of the chemotherapeutic agent affects the rapidly dividing crypt cells of the intestinal epithelium, and due to depletion of enzymes, leads to decreased absorption and most commonly observed increase in fluid retention [154,167]. The villi and mature cells of the intestinal wall are directly harmed, leading to a higher proportion of the immature secretory cells. The increase in fluid retention and decrease in the absorptive capacity of the villi cause an alteration in the osmotic gradient within the GIT that leads to the onset of diarrhea [154,168]. In addition to this mechanism, alteration of the GIT microflora induced by chemotherapeutic agents causes an increase in solutes and chloride ions in the intestinal tracts [168].

The presence of chloride ions triggers a shift in the osmotic gradient, causing water to be retained in the gastric canal, which then causes osmotic diarrhea [167,169]. However, it is unclear whether all CID from chemotherapeutic agents is a result of alteration in the epithelial surface of the intestines [154]. Attribution has also been made to the enteric nervous damage caused by the chemotherapeutic agent, which might be responsible for gastrointestinal secretory and motility disturbances associated with the pathophysiology of CID [154,170,171]. Aside from the pathophysiological mechanisms of CID highlighted above, McQuade et al [172] identified mucositis, changes in intestinal microbiota, disruption in water and electrolyte balance within the GI, and mucosal inflammation along the GI tract, and chemotherapy-induced damage to the enteric nervous system (ENS) as major pathophysiological changes associated with CID in colorectal cancer [172].

The recommendation by the consensus conference on the management of CID for the management of uncomplicated CID is through modification of diet and the use of standard doses of medication like loperamide, octreotide, and tincture of opium in outpatient cases. Nonetheless, complicated diarrhea might require an aggressive approach and hospitalization, involving patients placed on anti-diarrhea medications and intravenous fluids [161,169].

#### 4.4.2. Cardiovascular Complications

Cardiovascular complications in colorectal cancer often develop perioperatively or during chemotherapy [173]. Out of 90% of adverse effects experienced by the patients during the first cycle of treatment, 39% have been reported to be cardiovascular (CV) events [173]. This CV includes hypertension, myocardial infarction, angina, arrhythmias, thrombotic events, heart failure, and in extreme cases, death [173].

5-FU has been reported to elicit 1.2% to 18% cardiotoxic events [174], with the most significant cardiotoxic presentation being angina pectoris and vasospastic angina [174,175], followed by other manifestations like dyspnea, palpitations, and hypotension [176]. A four-step approach has been proposed for managing 5-FU cardiotoxicity as follows: (i) Immediately discontinuing the 5-FU; (ii) applying practical symptomatic treatment (iii) confirming that the symptoms are related to 5-FU; and (iv) rechallenging with pharmacological prophylaxis or introducing a different chemotherapeutic regimen [173]. Initiation of calcium channel blockers or nitrates is the lone proposed treatment by the European Society of Cardiology in treating the fatal cardiotoxic consequences of 5-FU [177].

Capecitabine, an oral pro-drug of 5-FU, produces cardiotoxic events like angina-like chest pain, acute ischemic events [178], arrhythmias, heart failures [179], ventricular fibrillation [180], and sudden cardiac death linked to coronary vasospasm [181]. The topmost cardiotoxic events with capecitabine occur when in combination with oxaliplatin, and bevacizumab [175,181]. Treatment of this is often symptomatic [182].

Bevacizumab has been associated with hypersensitivity adverse reactions with dyspnea, hypotension, hypoxia, and fever resulting from the massive release of cytokines [173]. Another class effect associated with bevacizumab is hypertension and heart failure [183]. Hypertension is a reversible occurrence that is dependent on the duration of the drug use and the dose [184]. Thus, regular monitoring of hypertension is needed during administration of the dose [185]. The treatment proposed includes angiotensin-converting enzyme inhibitors (ACEi) or angiotensin receptor blockers (ARBs) [185]. Of great interest is that the induction of hypertension in patients using bevacizumab has been used as a predicting factor for antitumor efficacy and better survival [186]. Finally, bevacizumab also induced venous thrombotic events [187] but can be treated using oral anticoagulants [188].

Cetuximab produces complications like urticaria and bronchospasm that come with hypotension and, in extreme cases, angina, myocardial infarction, heart failure, shock, and sudden death [189]. Venous thromboembolism is also associated with cetuximab, with limited data [190]. Panitumumab, on the other hand, has cardiotoxic effects like hypotension, hypertension, and venous thromboembolic events. The cardiotoxic effects requiring attention in the two drugs are palpitation, chest pain, arrhythmias, and dyspnea [191,192].

#### 4.4.3. Neuropathic Complications

Chemotherapy-induced peripheral neuropathy (CIPN) is a frequent dose-limiting complication suffered by colorectal patients on chemotherapy. It is reported that about 30–40% of patients undergoing chemotherapy may develop CIPN and the severity is variable between patients [193]. The classes of drugs causing neurotoxicity include platinum, vinca alkaloids, and taxanes [193]. Symptoms associated with CIPN often begin during the first two months of treatment [193] and can stabilize soon after the completion of treatment. However, acute neurotoxicity may present with paclitaxel and oxaliplatin or a coasting effect associated with the discontinuation of cisplatin [193]. The manifestation of CIPN is an implication for a dose decrease or in some instances, stopping the chemotherapeutic drug [193]. Discontinuing the chemotherapeutic drug may hold back treatment [193,194].

CIPN is initiated by the neurotoxic effects of chemotherapeutic drugs on neurons, and the associated sensory symptoms are higher compared to motor or autonomic symptoms [193]. In the majority of patients, there is dose-dependent development of CIPN after several cycles of administration of the chemotherapeutic drugs [193]. The signs and symptoms associated with CIPN often arise due to the damaging effect of chemotherapeutic drugs on the dorsal root ganglion neurons or their axons This leads to “acral pain, sensory loss, and sometimes, sensory ataxia” [193]. Platinum compounds, including oxaliplatin which is widely used in colorectal cancer, cause sensory neuropathy [194]. This effect is attributed to the penetrability of the blood-nerve barrier at the point of the dorsal root ganglion. The platinum-based compounds triggered harm to the dorsal root ganglia neurons through the formation of adducts with nuclear and mitochondria DNA.

Thus, all platinum agents are known to cause long-term peripheral sensory damage attributed to neuropathy [195]. This phenomenon is seen in 30–40% of patients receiving oxaliplatin and cisplatin treatment [196]. Furthermore, oxaliplatin is associated with acute neuropathic pain, which contains cold-induced dysesthesia that is significant in the hands, face, and oral cavity. Therefore, intense pain may be induced by cold conditions like cold winds or cold drinks [193]. The ‘coasting’ phenomenon is a worsened CIPN for several months after the discontinuation of platinum chemotherapy (cisplatin and oxaliplatin) [193]. The mechanism for oxaliplatin-induced peripheral neurotoxicity (OIPN) is linked to ‘DNA damage, dysfunction of voltage-gated ion channels, neuroinflammation, transporters, oxidative stress, and mitochondrial dysfunction [194].

Presently, it has been difficult to develop CIPN-preventing agents because of the complications that surround such agents, decreasing the efficacy of the chemotherapeutic agent.

Another embraced approach is identifying patient-specific risk factors that can be translated to planning chemotherapy strategies for each patient [193]. These factors include dose, delivery route, concurrent medication usage, age, pre-existing neuropathy, and pre-existing conditions like diabetes [197]. A further approach involves identifying individual patient-specific mechanisms [198]. The only pharmacological treatment recommendation by the American Society of Clinical Oncology (ASCO) in treating CIPN is duloxetine [199]. Duloxetine is a serotonin-norepinephrine reuptake inhibitor (SNRI) that shows a substantial enhancement in pain management.

Other intervention modalities include the use of acupuncture, cryotherapy and compression therapy, surgical treatment, exercise, and diets [199]. The emerging therapy includes the use of botulinum toxin injections, ganglioside-monosialic acid, mitochondria enzyme, and immunomodulation [199,200,201,202,203]. For OIPN, the strategies for management include the use of ‘chemoprotectants (e.g., glutathione, Ca/Mg, ibudilast, etc.), dose reduction, chronomodulated infusion, reintroduction of oxaliplatin and topical administration, pressurized intraperitoneal aerosol chemotherapy (PIPAC), and hyperthermic intraperitoneal chemotherapy (HIPEC)’ [194].

### 4.5. Complications Associated with Targeted Therapy and Management

This class of therapeutics has delivered on the promise of target selectivity. However, they have also been associated with several toxicities that are not typically seen with traditional cytotoxic agents. The mechanisms underlying these novel toxicities have not yet been fully elucidated [204].

While small molecule inhibitors targeting the intracellular EGFR TK domain have demonstrated activity in other malignancies, only antibodies directed at the ligand-binding domain have shown efficacy in colorectal cancer. Currently, cetuximab is the only EGFR monoclonal antibody approved by the Food and Drug Administration (FDA) for the treatment of colorectal cancer, with others, such as panitumumab and matuzumab, undergoing clinical development.

Cetuximab is an IgG1 chimeric version of the murine monoclonal antibody M225 [205]. A phase I study identified skin rash as the primary toxicity and reported saturable clearance consistent with receptor binding. Overall, cetuximab treatment is generally well tolerated. Common side effects include fatigue and dermatologic reactions, while less frequent adverse effects, such as allergic reactions and electrolyte imbalances, have also been reported [206]. Typical management includes corticosteroids, epinephrine, oxygen, and antihistamines. Severe reactions, such as angioedema, require discontinuation of cetuximab and are best managed with epinephrine and supportive care [207].

The adverse event profile of panitumumab is similar to that of cetuximab. However, as panitumumab is a fully human antibody without a murine component, allergic reactions are uncommon (with only one possible hypersensitivity reaction reported). Additionally, cutaneous toxicity is almost universal in patients receiving panitumumab, which may be attributed to its higher affinity for the receptor [208,209]. The addition of bevacizumab, an antibody targeting VEGF to chemotherapy, has significantly improved survival in patients with metastatic colon cancer [210]. While generally well tolerated, bevacizumab has a unique side effect profile associated with vascular disturbances. Vascular development plays a crucial role in both cancer biology and normal physiological processes [211].

The toxicities associated with bevacizumab have been extensively documented across multiple trials. The most common side effects include hypertension and proteinuria. Less frequent but more severe complications, such as bowel perforation, arterial thrombotic events, and bleeding, have also been observed. These adverse effects do not appear to be strongly dose-dependent, as both 5 mg/kg and 10 mg/kg doses have been used in colorectal cancer trials. Most of these adverse effects can be managed symptomatically [210,212].

### 4.6. Complications Associated with Immunotherapy and Management

While immunotherapeutic agents are relatively well tolerated, their uses have been linked to a range of toxicities and adverse effects which may be easily contained in most cases or severe and fatal in rare cases and may ultimately lead to therapy discontinuation [213,214]. The most common immunotherapy-related adverse effects are dermatological, gastrointestinal, hepatic, and endocrine toxicities [215]. PD-1 inhibitors, including nivolumab and pembrolizumab, exhibit a reduced occurrence of side effects compared to CTLA-4 blockers like ipilimumab [216]. The concomitant use of two or more immunotherapeutics or a combination with other treatment modalities such as chemotherapy and radiotherapy may increase the incidence, severity, and onset of these unwanted adverse effects. For example, a higher incidence of immune-related adverse drug events was observed in a combination of nivolumab and ipilimumab. In contrast to 20% of patients treated with nivolumab alone, over 30% of patients who received combination therapy experienced grade 3–4 treatment-related adverse events [216,217].

Skin rash and pruritus are very common among patients receiving immunotherapy and the management varies depending on the severity. Grade 1 and 2 can be managed with topical corticosteroid in creams while grade 3 and 4 require a more serious measure like high-dose steroids or discontinuation of therapy [218]. About one in three colorectal cancer patients receiving immune checkpoint inhibitor ipilimumab have diarrhea [219]. The incidence rate rises by over 10% when combined with nivolumab. Having ruled out other potential causes of diarrhea, treatment options range from antidiarrheal medications to systemic corticosteroids such as methylprednisolone and budesonide, and withdrawing the therapeutic agents causing the toxicity, depending on the severity of the condition [219]. Infliximab, a tumor necrosis factor-alpha inhibitor, and vedolizumab, an integrin receptor antagonist, are other safe and efficient alternatives used in severe diarrhea and colitis [219,220]. The use of steroids and immunosuppressive therapy to mitigate adverse toxicities has the propensity to diminish the anti-cancer effect of immunotherapeutics. As a result, the dosage and timing of these agents should be carefully considered. Steroid use soon after immunotherapy is discouraged whenever possible, and low-dose steroids are advised.

Hypothyroidism, hyperthyroidism, type 1 diabetes mellitus, and hypophysitis (pituitary inflammation) are some of the immunotherapy-related adverse effects of the endocrine [213,221]. Although less common, they are serious and life-threatening adverse effects. Hypothyroidism is often managed with beta-blockers to relieve presented symptoms or by hormonal replacement (levothyroxine) in severe cases of the side effects [221]. Other less common adverse effects are hepatitis, pneumonitis, infusion reaction, anemia, myocarditis, among others [213,216].

## 5. Conclusions

Treatment strategies for CRC continue to evolve rapidly. Presently, the focus is tilting towards personalized medicine, robot-assisted surgery, and laparoscopic procedures for minimal invasion during surgery. Other innovative approaches include discoveries in immunotherapy and targeted therapies, and the use of combination therapeutic approaches. These approaches hold great promise for more effective solutions to CRC treatment. However, complications arising from available treatment approaches remain challenging.

Despite the successes recorded earlier on the introduction of targeted therapies, limitations based on tumor heterogeneity, drug resistance, and toxicity concerns appear very soon. Clinical reports have shown that anti-EGFR therapies—cetuximab, panitumumab—were ineffective in treating patients with RAS-mutated tumors, thus confining their use to patients with RAS wild-type tumors. In addition, BRAF inhibitors have shown limited therapeutic efficacy when used as monotherapy, creating a need for combination therapy with a potential increase in risk of toxicity. Furthermore, although tyrosine kinase inhibitors showed high efficacy in late-stage CRC, high toxicity and the associated complications of hypertension and severe fatigue have limited their clinical utility.

Therefore, as efforts are being advanced toward developing innovative approaches to CRC treatment, the knowledge of existing complications associated with current therapies should help provide platforms for anticipating and proactively developing strategies for mitigating potential complications. Future initiatives should center around overcoming drug resistance via various mechanisms, developing personalized medicines, and optimizing combination strategies to ensure therapeutic efficacy while minimizing adverse effects. Having a deeper understanding of the tumor microenvironment and identifying novel biomarkers could ensure more precision and effective treatment strategies that address the current limitations in CRC therapy.

These will enable usability of developed solutions in ways that are consistent with adverse events minimization, with potential treatment tolerability, and resulting patient compliance for effective therapeutic outcomes. Using the knowledge of the molecular mechanisms of CRC development and the procedures for ADE manifestation would enable bringing more patient-friendly solutions to clinical practice. As usual, a multidisciplinary approach involving cancer biologists, molecular chemists, and formulation scientists among others, would be needed to achieve this goal.

## Figures and Tables

**Figure 1 biomedicines-13-01646-f001:**
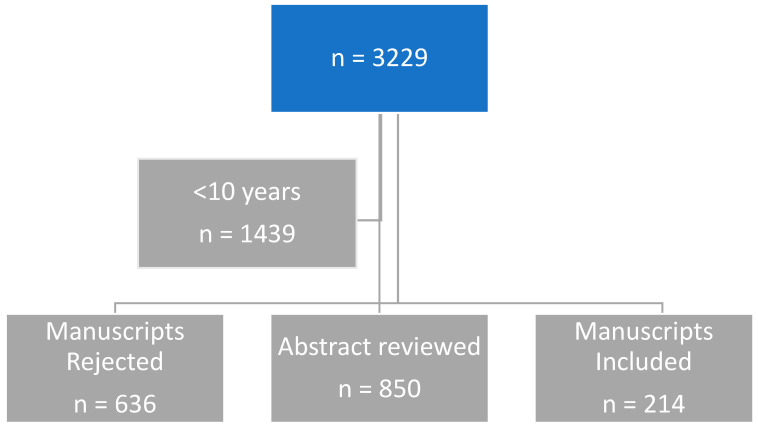
Search terms and articles found in PubMed search.

**Figure 2 biomedicines-13-01646-f002:**
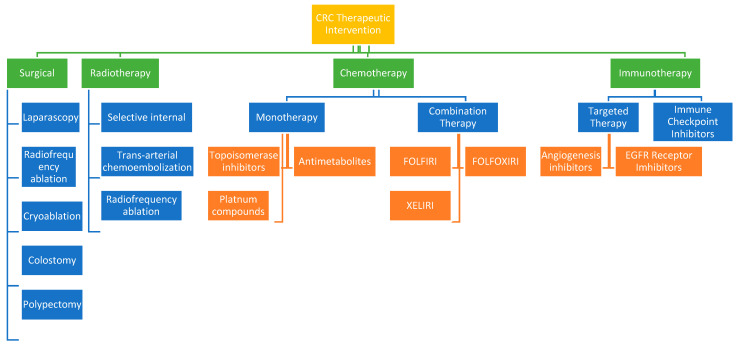
Summary of surgical and therapeutic interventions with colorectal cancer. FOLFIRI contains Irinotecan, 5-FU, and Leucovorin; FOLFOXIRI contains Irinotecan, Platinum compound (Oxaliplatin), and 5-FU/Leucovorin; XELIRI contains Irinotecan and Capecitabine.

**Figure 3 biomedicines-13-01646-f003:**
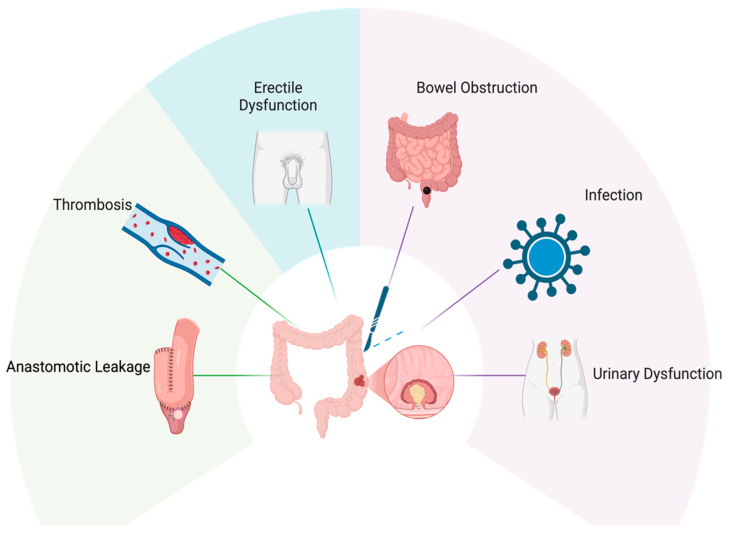
Complications associated with surgical intervention in CRC treatment.

**Figure 4 biomedicines-13-01646-f004:**
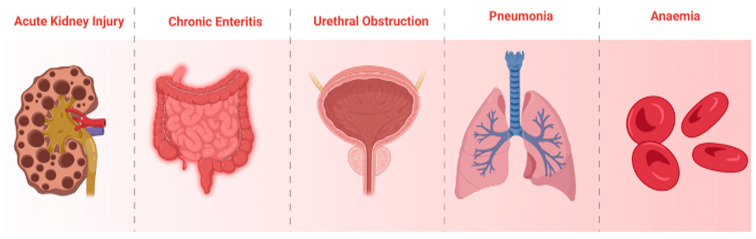
Complications associated with radiotherapy interventions in CRC treatment.

**Figure 5 biomedicines-13-01646-f005:**
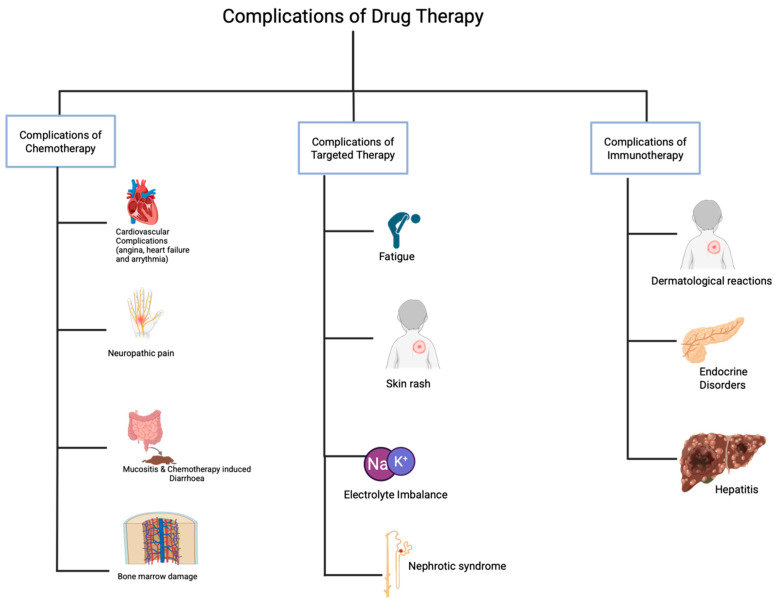
Complications associated with drug interventions in colorectal cancer treatment.

## Data Availability

The data presented in this study are available on request from the corresponding author due to related studies that are ongoing and pending patent applications.

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
