# Peer review of "Colorectal Cancer: Therapeutic Approaches and Their Complications"

_biomedicines, 2025, doi:10.3390/biomedicines13071646_

Round 1

Reviewer 1 Report

Comments and Suggestions for Authors

Adebayo, et al report a review article that descript of the therapeutic approaches and their complications in colorectal cancer treatment. Because it is a review article, some points need to be more clear and comprehensive.

  1. 4.1. Fluoropyrimidine, you mention about the mechanism of 5FU is inhibiting DNA replication. However, depend on the infusion rate, 5FU can also work on RNA level. Please review more article about that.
  2. You cited many information from other review article. It may be better to cite the original article. For example, the ref 19 on line 291, you may cite NEJM, 2007; 357:2040-8 (that the original information from).
  3. 2. complications associated with radiotherapy and management. Line 554, The figure should be Fir 4, not Fig 2. On the other hand, the treatment group in ref 123 were chemotherapy with or without radiotherapy. The side effects from that reference should not put in this paragraph. Those may be more likely related to chemotherapy, including neutropenia, sepsis, anemia, etc.
  4. Line 580~583, you mentioned about the infertility risk of female CRC patients. In my knowledge, it should related to “target area”, like sigmoid, rectum or anus, those nearby ovaries. Is that correct? If that is true, please mention more about the target area of colon that radiotherapy designed in those references. Do not just use CRC that including cecum, ascending, transverse, descending, sigmoid colon and rectum.
  5. Line 603, 604, the authors did not mention about oxaliplatin in ref. 140, and 141.
  6. Figure 5, there are still some other common side effects, like bone marrow and mucosa damage from chemotherapy, nephrotic syndrome, hypertension from targeted therapy, etc. please make if more comprehensive.
  7. 3.2 Cardiovascular complications, line 646, normally, the CV complications during chemotherapy are “hypotension”, myocardial infarction, etc. Hypertension may related to tyrosine kinase inhibitor, and VEGF/VEGFR antagonists. Please correct it.
  8. Line 688, “Discontinuing the chemotherapeutic drug may hold back treatment” is confuse to me. It is difficult to understand what do you want to let readers know.
  9. Line 734-738, a small molecule inhibitor, regorafenib (stivarga), which target on multiple tyrosine kinases, was approved by FDA in CRC. Please mention that.
  10. Mucositis is more common than electrolyte imbalances, but less common than skin reactions during EGFR antagonists treatment. I do not see that in your manuscript.
  11. Line 779, skin rash and pruritus are very common among patients receiving immunotherapy… How common (frequency) is very common?

Author Response

Reviewer's comment 1: Fluoropyrimidine, you mention about the mechanism of 5FU is inhibiting DNA replication. However, depend on the infusion rate, 5FU can also work on RNA level. Please review more article about that.

RESPONSE: Addressed . Please see line 170 to 180

Reviewer's comment 2: You cited many information from other review article. It may be better to cite the original article. For example, the ref 19 on line 291, you may cite NEJM, 2007; 357:2040-8 (that the original information from).

RESPONSE: Requested changes have been made in the ref 19 and throughut the manuscript.

Reviewer's comment 3: complications associated with radiotherapy and management. Line 554, The figure should be Fir 4, not Fig 2. On the other hand, the treatment group in ref 123 were chemotherapy with or without radiotherapy. The side effects from that reference should not put in this paragraph. Those may be more likely related to chemotherapy, including neutropenia, sepsis, anemia

RESPONSE: Fig number has been changed appropriately. The side effect has been matched to those of chemotherapy as suggested.

Reviewer's comment 4: Line 580~583, you mentioned about the infertility risk of female CRC patients. In my knowledge, it should related to “target area”, like sigmoid, rectum or anus, those nearby ovaries. Is that correct? If that is true, please mention more about the target area of colon that radiotherapy designed in those references. Do not just use CRC that including cecum, ascending, transverse, descending, sigmoid colon and rectum

RESPONSE: Yes, that is perfectly correct. Contents of lines 580~583 have been rewritten into lines 684 to 696 to address this concern, with additional refences.

Reviewer's comment 5: Line 603, 604, the authors did not mention about oxaliplatin in ref. 140, and 141.

RESPONSE: Oxiplatin is categorzied under platinum compounds in Fig. 2. It has been added to the legend to the figure. Further discussion on oxiplatin can be found in llines 849 to 864.

Reviewer's comment 6: Figure 5, there are still some other common side effects, like bone marrow and mucosa damage from chemotherapy, nephrotic syndrome, hypertension from targeted therapy, etc. please make if more comprehensive.

RESPONSE: Agree. The suggested side effects have been included in the figure.

Reviewer's comment 7: Cardiovascular complications, line 646, normally, the CV complications during chemotherapy are “hypotension”, myocardial infarction, etc. Hypertension may related to tyrosine kinase inhibitor, and VEGF/VEGFR antagonists. Please correct it.

RESPONSE: The section has been revised for accuracy and fluidity. Most literature refrenced Bevacizumab, Cetuximab and Panitumumab with "hypotension". Those connected with hypertension are anthracyclines (Doxorubicin); kinase inhibitor (Sultinib), VEGF-A antagonist (Aflibercept), and FDA-approved humanized MAb (Bevacizumab and Sultinib). These CV complications by therapeutic agents have been discussed in detail in lines 785 to 828.

Reviewer's comment 8:Line 688, “Discontinuing the chemotherapeutic drug may hold back treatment” is confuse to me. It is difficult to understand what do you want to let readers know.

RESPONSE: Yes, in a life-threathening condition caused by a chemotherapy, the first step to relieving the patient would be to discontinue that offending therapeutic agent. This is usually done while concurrently initiating an alternative therapy as indicated in step iv of the process. 

Reviewer's comment 9: Line 734-738, a small molecule inhibitor, regorafenib (stivarga), which target on multiple tyrosine kinases, was approved by FDA in CRC. Please mention that.

RESPONSE: Agree, incorporated. Please see lines 351 to 355. Regorafenib (Fluror-sorafenib, stivarga) has a chemical structure that is closely related to sorafenib and is the first approved pharmacotherapy for mCRC patients who are non-responsive to current standard therapies. 

Reviewer's comment 10: Mucositis is more common than electrolyte imbalances, but less common than skin reactions during EGFR antagonists treatment. I do not see that in your manuscript.

RESPONSE: Agree. This has been incorporated in section 3.5 lines 274 to 281. Mucositis is further discussed in Section 4.4.1 Chemotherapy-induced diarhea lines 751 - 777.

Reviewer's comment 11: Line 779, skin rash and pruritus are very common among patients receiving immunotherapy… How common (frequency) is very common?

RESPONSE: This has been addressed. Please see section 4.5, lines 938 to 954.

Reviewer 2 Report

Comments and Suggestions for Authors

1.This review paper provides a comprehensive summary of standard treatment methods for colorectal cancer (CRC), associated complications, and explores the progress in novel therapeutic approaches. It is recommended to include a more in-depth discussion on "personalized therapy," such as treatment selection based on biomarkers, and to supplement the analysis with insights into resistance mechanisms and strategies to overcome them. This is also a current research hotspot in CRC.
2.The paper is generally scientifically sound and logically structured, with reliable data sources and conclusions based on existing literature and clinical research. However, the following points require attention:
Literature Timeliness: Some cited references are outdated (e.g., studies from the early 2000s). It is advised to update them to literature from the past 3-5 years that reflects key advancements in the field.
Descriptive Prudence: Descriptions of certain therapeutic modalities (e.g., the efficacy of immunotherapy) should be more cautious, and their applicable patient populations need to be clearly specified.
Mechanistic Elaboration: The pathophysiological mechanisms underlying some complications (e.g., chemotherapy-induced diarrhea) could be further elaborated. It is recommended to cite more relevant basic research literature for support to enhance the depth of the discussion.
3.In the "Conclusion" section, the limitations of current therapeutic approaches should be clearly articulated. Additionally, supplementing the discussion with examples of negative findings (e.g., failure cases of certain targeted drugs) would help demonstrate academic rigor.

Author Response

Comment 1: This review paper provides a comprehensive summary of standard treatment methods for colorectal cancer (CRC), associated complications, and explores the progress in novel therapeutic approaches. It is recommended to include a more in-depth discussion on "personalized therapy," such as treatment selection based on biomarkers, and to supplement the analysis with insights into resistance mechanisms and strategies to overcome them. This is also a current research hotspot in CRC.

Response 1: Agree. A whole section 3.8 lines 502 to 538 has been dedicated to personalized therapy.

Comment 2: The paper is generally scientifically sound and logically structured, with reliable data sources and conclusions based on existing literature and clinical research. However, the following points require attention:
Comment 2.1: Literature Timeliness: Some cited references are outdated (e.g., studies from the early 2000s). It is advised to update them to literature from the past 3-5 years that reflects key advancements in the field.

Response 2.1: Agree. The core of the article centers on the previous 10 years' publications on the subject as using the "Search terms" and articles found in PubMed as primary source. The manuscript has been enriched with more recent literature where available. Also, the evolutionary trend of CRC development, therapeutic approaches and innovations transcend last 5 years period. In order to include significant milestones in the development of therapeutic approaches, older literatures would have to be iincluded as is done in this manuscript.

Comment 2.2: Descriptive Prudence: Descriptions of certain therapeutic modalities (e.g., the efficacy of immunotherapy) should be more cautious, and their applicable patient populations need to be clearly specified.

Response 2.2: Agree. We have clearly indicated that small molecule inhibitors targeting the intracellular EGFR TK domain have demonstrated activity in other malignancies, but only antibodies directed at the ligand-binding domain have shown efficacy in colorectal cancer. We highlighted that cetuximab is the only EGFR monoclonal antibody approved by the US FDA for the treatment of CRC and that others (such as panitumumab and matuzumab) are still undergoing clinical development. The therapeutic benefits, adverse common effects and possible methods for their mitigation have been incorporated in various sections of the manuscript, specifically in lines 804-809; 823 – 830 and 859 to 864.

Comment 2.3: Mechanistic Elaboration: The pathophysiological mechanisms underlying some complications (e.g., chemotherapy-induced diarrhea) could be further elaborated. It is recommended to cite more relevant basic research literature for support to enhance the depth of the discussion.

Response 2.3: Agree. Elaboration on the pathophysiologic mechanisms has been added to section 4.4.1 line 708 to 712.

Comment 3: In the "Conclusion" section, the limitations of current therapeutic approaches should be clearly articulated. Additionally, supplementing the discussion with examples of negative findings (e.g., failure cases of certain targeted drugs) would help demonstrate academic rigor.

Response 3: Agree. please see conclusion enrichment in lines 893 to 925